# FILTER PRE-PRUNING FOR IMPROVED FINE-TUNING OF QUANTIZED DEEP NEURAL NETWORKS

## ABSTRACT

Deep Neural Networks(DNNs) have many parameters and activation data, and these both are expensive to implement. One method to reduce the size of the DNN is to quantize the pre-trained model by using a low-bit expression for weights and activations, using fine-tuning to recover the drop in accuracy. However, it is generally difficult to train neural networks which use low-bit expressions. One reason is that the weights in the middle layer of the DNN have a wide dynamic range and so when quantizing the wide dynamic range into a few bits, the step size becomes large, which leads to a large quantization error and finally a large degradation in accuracy. To solve this problem, this paper makes the following three contributions without using any additional learning parameters and hyper-parameters. First, we analyze how batch normalization, which causes the afore-mentioned problem, disturbs the fine-tuning of the quantized DNN. Second, based on these results, we propose a new pruning method called Pruning for Quantization (PfQ) which removes the filters that disturb the fine-tuning of the DNN while not affecting the inferred result as far as possible. Third, we propose a work-flow of fine-tuning for quantized DNNs using the proposed pruning method(PfQ). Experiments using well-known models and datasets confirmed that the proposed method achieves higher performance with a similar model size than conventional quantization methods including fine-tuning.

## 1 INTRODUCTION

DNNs (Deep Neural Networks) greatly contribute to performance improvement in various tasks and their implementation in edge devices is required. On the other hand, a typical DNN (He et al., 2015; Simonyan & Zisserman, 2015) has the problem that the implementation cost is very large, and it is difficult to operate it on an edge device with limited resources. One approach to this problem is to reduce the implementation cost by quantizing the activations and weights in the DNN.

Quantizing a DNN using extremely low bits, such as 1 or 2 bits has been studied by Courbariaux et al. (2015) and Gu et al. (2019). However, it is known that while such a bit re-duction has been performed for a large model such as ResNet (He et al., 2015), it has not yet been performed for a small model such as MobileNet (Howard et al., 2017; Sandler et al., 2018; Howard et al., 2019), and is very difficult to apply to this case. For models that are difficult to quan-tize, special processing for the DNN is required before quantization. On the other hand, although fine-tuning is essential for quantization with extremely low bit representation, few studies have been conducted on pre-processing for easy fine-tuning of quantized DNNs. In particular, it has been ex-perimentally shown from previous works (Lan et al., 2019; Frankle & Carbin, 2018) that, regardless of quantization, some weights are unnecessary for learning or in fact disturb the learning process. Therefore, we focused on the possibility of the existence of weights that specially disturb the fine-tuning quantized DNN and to improve the performance of the quantized DNN after fine-tuning by removing those weights.

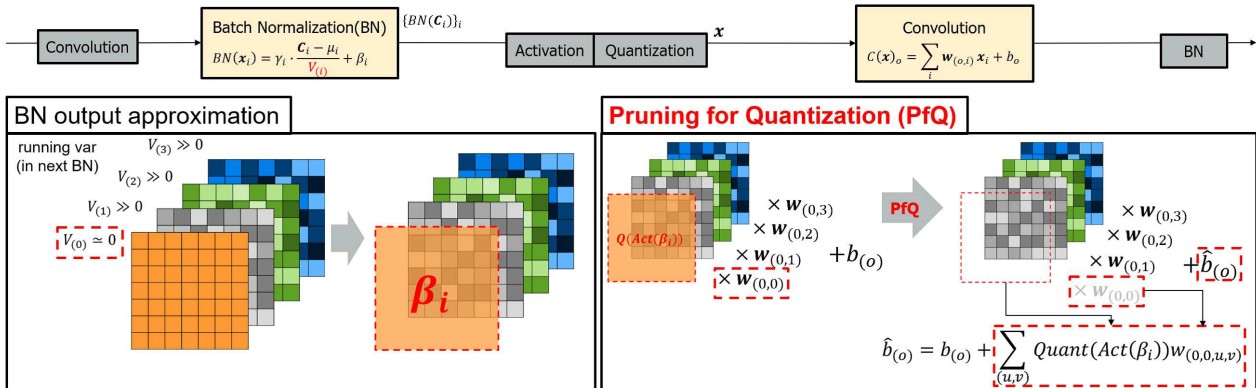

Figure 1: This is a diagram of the proposed pruning method PfQ. As mentioned in subsection 3.1, the filters with a very small running variance can approximate its output by $\beta_i$. At this time, by pruning the filters and correcting the bias of the next convolution using $\beta_i$ as described above, the DNN can be pruned without affecting the output as far as possible.

## 2 RELATED WORK AND PROBLEM

### 2.1 CONVENTIONAL QUANTIZATION WORKS

In order to reduce the implementation cost of DNN in hardware, many methods of reduction by pruning (Molchanov et al., 2019; You et al., 2019; He et al., 2019) and quantization (Gu et al., 2019; Gong et al., 2020; Fan et al., 2020; Uhlich et al., 2019; Sheng et al., 2018; Nagel et al., 2019) have been studied. Research is also underway in network architectures such as depthwise convolution and pointwise convolution (Howard et al., 2017), which are arithmetic modules that maintain high performance even when both the number of operations and the model capacity are kept small (Howard et al., 2017; Sandler et al., 2018; Howard et al., 2019). Knowledge Transfer (Hinton et al., 2015; Zagoruyko & Komodakis, 2016; Radosavovic et al., 2018; Chen et al., 2019) is also being studied as a special learning method for achieving high performance in these compression technologies and compressed architectures.

There have been various approaches to studies in DNN compression using quantization. DNN quantization is usually done for activations and weights in DNN. As a basic approach to quantization, there are some methods for a pre-trained model of which one uses only quantization (Sheng et al., 2018; Nagel et al., 2019) and another uses fine-tuning (Gong et al., 2019; Gu et al., 2019; Jung et al., 2019; Uhlich et al., 2019; Gong et al., 2020; Fan et al., 2020). Cardinaux et al. (2020) and Han et al. (2015) propose a method for expressing values that are as precise as possible. However, the simplest expression method is easier to calculate and implement in hardware, so expressions to divide the range of quantized values by a linear value (Jacob et al., 2018; Sheng et al., 2018; Nagel et al., 2019; Gu et al., 2019; Gong et al., 2019; 2020; Jung et al., 2019) or a power of 2 value (Lee et al., 2017; Li et al., 2019) are widely used at present in quantization research.

It is known that fine-tuning is essential to quantize DNN with low bits in order not to decrease accuracy as much as possible. Jacob et al. (2018) shows that an accuracy improvement can be expected by fine-tuning with quantization. Since the quantization function is usually non-differentiable, the gradient in the quantization function is often approximated by STE (Bengio et al., 2013). However, in the case of quantization with low bits, the error is very large in the approximation by STE, and it is known that it is difficult to accurately propagate the gradient backward from the output side to the input side, making learning difficult. In order to solve this problem, Gong et al. (2019), Fan et al. (2020) and Darabi et al. (2018) proposed the backward quantization function so that the gradient can be propagated to the input side as accurately as possible. Gu et al. (2019) and Banner et al. (2018) considered activations and weights in DNN as random variables, and proposed an approach to optimize the stochastic model behind DNN so that activations and weights in DNN become vectors which are easy to quantize. Jung et al. (2019) and Uhlich et al. (2019) considered that it is important

for performance to determine the dynamic range of vectors to be quantized appropriately, and it is proposed that this dynamic range is also learned as a learning parameter.

## 2.2 PROBLEM OF WIDE DYNAMIC RANGE ON QUANTIZATION

It is important to set the dynamic range appropriately in the quantization. In modern network architectures, most of the computation is made up of a block of convolution and batch normalization (BN). It is known that the above block can be replaced by the equivalent convolution, and quantization is performed after this replacement. Thus, it is possible to suppress the quantization error rather than quantizing the convolution and BN separately. On the other hand, it is also known that the dynamic range of the convolution may be increased by this replacement (Sheng et al., 2018; Nagel et al., 2019). In order to explain this in detail, the calculation formulas of convolution and BN in pre-training are described. First, the equation for convolution is described as follows

$$C_{(o,j,k)} = \sum_i \sum_{(u,v)} w_{(o,i,u,v)} x_{(i,j+u,k+v)} + b_{(o)}, \tag{1}$$

where $C_{(o,j,k)}$ is the $o$-th convolution output feature at the coordinate $(j, k)$, $w_{(o,i,u,v)}$ is the weight of the kernel coordinates $(u, v)$ in the $i$-th input map, and $x_{(i,j+u,k+v)}$ is the input data. For simplicity, $C_{(o,j,k)}$ is represented as vector $\mathbf{C}_{(o)} = (c_{(o,1)}, ..., c_{(o,N)})$[1] and $w_{(o,i,u,v)}$ is represented as vector $\mathbf{w}_{(o,i)}$ if it is not necessary. Next, the calculation formula for BN is described as

$$\mathbf{B}_{(o)} = \frac{\mathbf{C}_{(o)} - \mu_{(o)}}{\sqrt{(\sigma_{(o)})^2 + \epsilon}} \gamma_{(o)} + \beta_{(o)}, \tag{2}$$

where $\mathbf{B}_{(o)}$ is the $o$-th BN output feature[2]. $\mu_{(o)}, (\sigma_{(o)})^2$ are respectively calculated by equation (3), (4) during the training or by equation (7), (8) during the inference. $\gamma_{(o)}, \beta_{(o)}$ are respectively the scaling factor and shift amounts of BN. The mean and variance are calculated as

$$\mu_{(o)} = \frac{1}{N} \sum_n^N c_{(o,n)} \tag{3}$$

$$(\sigma_{(o)})^2 = \frac{1}{N} \sum_n^N \left| c_{(o,n)} - \mu_{(o)} \right|^2. \tag{4}$$

At the time of inference or the fine-tuning of quantization for the pre-trained model, BN can be folded into convolution by inserting equation (1) into (2). The new weight ($\hat{w}_{(o,i,u,v)}$) and bias ($\hat{b}_{(o)}$) of the convolution when BN is folded in the convolution are calculated by

$$\hat{\mathbf{w}}_{(o,i)} = \frac{\gamma_{(o)}}{\sqrt{V_{(o)}^{(\tau)} + \epsilon}} \mathbf{w}_{(o,i)} \tag{5}$$

$$\hat{b}_{(o)} = \frac{\gamma_{(o)}}{\sqrt{V_{(o)}^{(\tau)} + \epsilon}} b_{(o)} + \beta_{(o)} - \frac{\gamma_{(o)}}{\sqrt{V_{(o)}^{(\tau)} + \epsilon}} M_{(o)}^{(\tau)}, \tag{6}$$

where $M_{(o)}^{(\tau)}, V_{(o)}^{(\tau)}$ are respectively the mean (running mean) and variance (running variance) of BN used at inference, and the values are calculated as

$$M_{(o)}^{(\tau)} = M_{(o)}^{(\tau-1)} \rho + \mu_{(o)}(1 - \rho) \tag{7}$$

$$V_{(o)}^{(\tau)} = V_{(o)}^{(\tau-1)} \rho + (\sigma_{(o)})^2 (1 - \rho) \frac{N}{N-1}, \tag{8}$$

where $\tau$ is an iteration index, and is omitted in this paper when not specifically needed, and $\rho$ is a hyper-parameter of BN in pre-training so that $0 < \rho < 1$. The initial values of (7) and (8) are set before learning in the same way as weights such as convolution.

Sheng et al. (2018) pointed out that the dynamic range of the weights of depthwise convolution is

---

[1] $N$ is the number of elements in a channel $\mathbf{C}_{(o)}$.

[2] Vector and scalar operations are performed by implicitly broadcasting scalars.

large, which leads to the large quantization error. They also pointed out the relationship that the filters (sets of weights) with wide dynamic range correspond to the running variances of the BN with small magnitude. Nagel et al. (2019) attacked this problem by adjusting scales of weights of consecutive depthwise and pointwise convolutions without affecting the outputs of the pointwise convolution. However, they cannot solve the problem in quantization completely.

In this study, we theoretically analyze the weights with the running variances which have small magnitude and show that certain weights disturb the fine-tuning of the quantized DNN. Based on this analysis, we propose a new quantization training method which can solve the problem and improve the performance (section 4).

## 3 PROPOSAL

### 3.1 WEIGHTS DISTURBING LEARNING

We analyze how the fine-tuning is affected when the following condition holds

$$V_{(c)}^{(L,\tau)} \simeq 0, \tag{9}$$

where $V_{(c)}^{(L,\tau)}$ is the running variance (8) of $c$th-channel in $L$th-BN. When (9) holds regarding a significantly large $\tau$, the term of the left side and the first term of the right side in (8) are close to zero. Therefore, it is necessary that the second term of the right side in (8) close to zero to hold (8). As the result, the variance $(\sigma_{(c)})^2 \simeq 0$ holds, the following equation

$$|C_{(c,n)}^{(L)} - \mu_{(c)}^{(L)}| \simeq 0 \tag{10}$$

holds for arbitrary $n \in \{1, ..., N\}$ and input data from (4). $\epsilon$ in (2) is usually set to satisfy $\epsilon \gg 0$. Therefore, in the case of $(\sigma_{(c)})^2 \simeq 0$, since $(\sigma_{(c)})^2 \ll \epsilon$ holds, the following equation

$$\mathbf{B}_{(c)}^{(L)} \simeq \beta_{(c)}^{(L)} \tag{11}$$

holds from (2) and (10). Since the output of BN is a constant value by (11) regardless of the input data, the following convolution can be calculated as

$$
\begin{aligned}
C_{(o,j,k)}^{(L+1)} & \simeq \sum_{\substack{i \\ i \neq c}} \sum_{(u,v)} w_{(o,i,u,v)}^{(L+1)} x_{(i,j+u,k+v)}^{(L+1)} + \sum_{(u,v)} w_{(o,c,u,v)}^{(L+1)} Act(\beta_{(c)}^{(L)}) + b_{(o)}^{(L+1)} \quad (12) \\
& = \sum_{\substack{i \\ i \neq c}} \sum_{(u,v)} w_{(o,i,u,v)}^{(L+1)} x_{(i,j+u,k+v)}^{(L+1)} + \hat{b}_{(o)}^{(L+1)} \quad (13)
\end{aligned}
$$

$$
\hat{b}_{(o)}^{(L+1)} = b_{(o)}^{(L+1)} + U(\mathbf{w}_{(o,c)}^{(L+1)}, \beta_{(c)}^{(L)}) \tag{14}
$$

$$
U(\mathbf{w}_{(o,c)}^{(L+1)}, \beta_{(c)}^{(L)}) = \sum_{(u,v)} w_{(o,c,u,v)}^{(L+1)} Act(\beta_{(c)}^{(L)}) \tag{15}
$$

where $Act(*)$ is the activation function. This is equivalent to the bias of the $L + 1$-th convolution being (14). The bias of (14) is updated as

$$
\begin{aligned}
\hat{b}_{(o)}^{(L+1,\tau)} = b_{(o)}^{(L+1,\tau-1)} & + U(\mathbf{w}_{(o,c)}^{(L+1,\tau-1)}, \beta_{(c)}^{(L,\tau-1)}) \\
& + \Delta b_{(o)}^{(L+1,\tau-1)} + \Delta U(\mathbf{w}_{(o,c)}^{(L+1,\tau-1)}, \beta_{(c)}^{(L,\tau-1)}) \quad (16)
\end{aligned}
$$

where $\tau$ is an index of the iteration to express the bias before and after the update. At the step of quantization, since each of the first and second terms of (14) is quantized, the quantization error tends to be larger than when the whole of (14) is quantized. Further, in addition to the quantization error of (14), equation (16) adds an approximation error in approximating the gradient of the quantization function to the third and fourth terms, respectively. Therefore, it can be seen that the quantization error of $\hat{b}_{(o)}^{(L+1)}$ tends to be larger than that of the other weights in the fine-tuning of the quantized DNN. From the above, it can be seen that the filters satisfying (9) disturb the fine-tuning.

### 3.2 Proposal of New Pruning Method to Remove Weights Disturbing Training

As mentioned in subsection 3.1, weights that satisfy $V_{(o)}^{(\tau)} \simeq 0$ extend the dynamic range and disturb the fine-tuning of the quantized DNN. Therefore, we propose a new pruning method, illustrated in Figure 1, called Pruning for Quantization (PfQ), that solves the described problem by removing the weights that satisfies the following conditional

$$V_{(o)}^{(L,\tau)} < \epsilon \tag{17}$$

for the running variance $V_{(o)}^{(L,\tau)}$ of the $L$-th BN, correcting the bias of the next convolution without affecting the output of the DNN as far as possible. The reason behind this requirement for pruning is that the term controlling the magnitude of the denominator becomes $\epsilon$ in (5) at this time, increasing the dynamic range of that weight.

If the convolution has a bias, we replace the value with the one calculated by (14). Thus, at the time of the fine-tuning of the quantized DNN, the second term in (16) is absorbed into the first term, and the fourth term disappears by pruning, thereby eliminating the bad effect on the fine-tuning discussed in subsection 3.1. When a BN is used, a convolution before the BN often does not have a bias in the convolution immediately before the BN. In this case, the $\beta$ of the BN is replaced by

$$\hat{\beta}_{(o)}^{(L+1)} = \beta_{(o)}^{(L+1)} + \frac{\gamma_{(o)}^{(L+1)}}{\sqrt{V_{(o)}^{(L+1,\tau)} + \epsilon}} \sum_{(u,v)} w_{(o,c,u,v)}^{(L+1)} Act(\beta_{(c)}^{(L)}). \tag{18}$$

When we quantize the activations, we use the quantized $Act(\beta_{(c)}^{(L)})$ in (15) and (18). The proposed pruning method was summarized in Figure 1.

### 3.3 Proposal of Quantization Workflow

Also, our quantization workflow can get the effect of BN. We propose a new quantization workflow (Algorithm 1) in this section. In general, DNNs are quantized simultaneously for activations and weights for the pre-trained model with float. However, the fine-tuning may not proceed well by this method since the variation from the pre-trained model is too large. Therefore, we fine-tune the quantized DNN gradually in the proposed quantization workflow. First, we fine-tune the DNN for quantized activations only, then for quantized activations and weights in the DNN. Furthermore, in general, BN is folded into the convolution when the weights are quantized. This does not allow us to get the effect from BN in the fine-tuning of the quantized DNN. On the other hand, our quantization workflow allows the first fine-tuning using BN without folding BN into the convolution since the fine-tuning of the DNN for quantized activations only is performed at first. By performing PfQ for the DNN before each fine-tuning, the problem claimed by (Sheng et al., 2018) that the dynamic range of the weights are widen and the problem claimed by subsection 3.1 in this paper that some biases tends to accumulate the quantization and approximation error are solved. Regarding the second PfQ, since BN still remains in the fine-tuning of the DNN for quantized activations, the fine-tuning may generate some filters with small variance. In order to remove these filters, the same processing as the first PfQ is performed.

## 4 Experiments

In this section, we experimentally confirmed the performance of the quantization method proposed in subsection 3.2 and 3.3. First, we explain the setup of the experiment. Next, in order to confirm the effect of the weights which disturb the fine-tuning, we compare the performance of the models whose activations are quantized and fine-tuned. The difference of these models is that some weights in the model are removed by PfQ or not. Finally, we compare the performance of our proposed quantization method with the other quantization methods.

---

**Algorithm 1** Proposal of Quantization Workflow

---

**Input:** pre-trained model, dataset, training hyperparameters (bitwidth, epochs $e_A, e_W$, the other training parameters)
**Output:** quantized model
  1: Perform PfQ operation to the pre-trained model
  2: **for** $i = 0$ to $e_A$ **do**
  3:    Fine-tune the pre-trained model for quantized activations
  4: **end for**
  5: Perform PfQ operation to the fine-tuned model for the quantized activations
  6: Fold BN
  7: **for** $i = 0$ to $e_W$ **do**
  8:    Fine-tune the model for the quantized activations and weights
  9: **end for**
 10: **return** quantized model

---

## 4.1 EXPERIMENTAL SETTINGS

We experiment with the classification task using CIFAR-100 and imagenet1000 as datasets. The learning rate was calculated by the following equation

$$l_e = \begin{cases} l_{-1} \times \left(1 + \cos\left(\dfrac{e - w}{\lambda}\pi\right)\right) & (e \geqq w) \\ l_{-1} \times \dfrac{e}{w} & (e < w), \end{cases} \tag{19}$$

where $e$ is the number of epochs, $l_{-1}$ is the initial learning rate, $w$ is the number of warm-up epochs, and $\lambda$ is the period of the cosine curve. The learning rate was reduced by a cosine curve (Loshchilov & Hutter, 2016) during $e \geq w$. In the following experiments, the proposed method learned 100 epochs in the each fine-tuning in Algorithm 1, respectively. We used DSQ (Gong et al., 2019), PACT (Choi et al., 2018; Wang et al., 2018) and DFQ (Nagel et al., 2019) as quantization methods to compare with our proposed method. The performance of DSQ and PACT are directly referenced from the values in the paper, but regarding DFQ, the performance of the fine-tuning of the quantized DNN was not described in the paper. Therefore the performance of DFQ was obtained by our own experiment with 200 epochs fine-tuning.

In this paper, we use the quantization function given by the following

$$Q(x) \quad = \quad round\left(\frac{min(max(x,m), M) - m}{scale}\right) \times scale + m \tag{20}$$

$$scale \quad = \quad \frac{M - m}{2^n} \tag{21}$$

where $x$ represents the input data (an activation or a weight) of the quantization function, $n$ represents the bit width, and $m, M$ represent the lower limit and upper limit of the quantization range, respectively. We use STE to calculate the gradient. In this paper, we use MobileNetV1 (Howard et al., 2017) and MobileNetV2 (Sandler et al., 2018) as the network architectures. At the time of quantization, for quantization of activations, all activations except the last layer and addition layers (i.e. outputs of skip connection) were quantized, and for weights, all convolutions, depthwise convolutions and affines including the first and last layers were quantized.

In this study, $\epsilon = 0.00001$ was used as $\epsilon$ in (17). This is the default value of the BN in NNabla (Sony).

## 4.2 ABLATION STUDY

### 4.2.1 EFFECT OF DISTURBING WEIGHTS

In this section, we made a comparative experiment to confirm that PfQ eliminates the bad effect on the fine-tuning of the quantized DNN described in subsection 3.1. In addition, in order to confirm the effect of the bias correction in PfQ, we made the experiment even in the case where the bias

correction was excluded. We made experiments for MobileNetV1 and MobileNetV2 for the classification task of CIFAR-100. In the experiment of CIFAR-100 in this paper, the GPU was single (2080Ti) and the batch size was 16. We made the pre-trained models of the two network architectures by the following experiment: the number of epochs was 600, the optimizer was momentum SGD, the moment was 0.9, the learning rate settings in (19) were $l_{-1} = 0.16, w = 0, \lambda = 40$, and the learning rate decay was not used. In the fine-tuning of the quantized DNN the following settings were used: the number of epochs was 100, the optimizer was the same as the pre trained model, the learning rate settings in (19) were $l_{-1} = 0.001, w = 0, \lambda = 100$, and the weight decay was 0.0001.

Table 1: Top1 Accuracy comparison between pruned and no pruned models

| Model | Activation bitwidth | Evaluation | original | PfQ (w/o bias correction) | PfQ |
|---|---|---|---|---|---|
| MobileNetV1 | 4 | Accuracy | 76.34 | 76.67 | **76.75** |
| | | MAC | 510M | **367M** | **367M** |
| MobileNetV1 | 3 | Accuracy | 74.17 | 75.14 | **75.26** |
| | | MAC | 510M | **367M** | **367M** |
| MobileNetV2 | 4 | Accuracy | 74.04 | 74.67 | **75.00** |
| | | MAC | 147M | **93M** | **93M** |
| MobileNetV2 | 3 | Accuracy | 71.36 | 72.17 | **72.28** |
| | | MAC | 147M | **93M** | **93M** |

The experimental results are summarized in Table 1. Regarding the accuracy in Table 1, Table 1 shows that the fine-tuning after using PfQ gives better performance in all experiments, and it can be solved that the bad effect on the fine-tuning of the quantized DNN described in 3.1. In particular, in PfQ (w/o bias correction), the accuracy is higher than that of the original, and the difference is larger for 3 bits than 4 bits, indicating that the influence of the quantization error (i.e. the 4th term in (16)) in the backward calculation is large. MobileNetV1 and MobileNetV2 in the columns of PfQ in Table 1 are pruned with the weights by 31.86% and 35.74%, respectively. Thus, the number of MACs is also reduced.

### 4.2.2 SOLVING DYNAMIC RANGE PROBLEM

In this section, we confirmed that PfQ can suppress the increase in the dynamic range of the weights after folding BN. The models that we used are MobileNetV1 and MobileNetV2 used in Table 1.

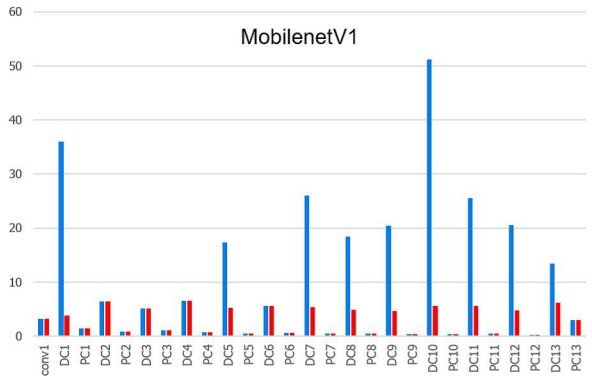 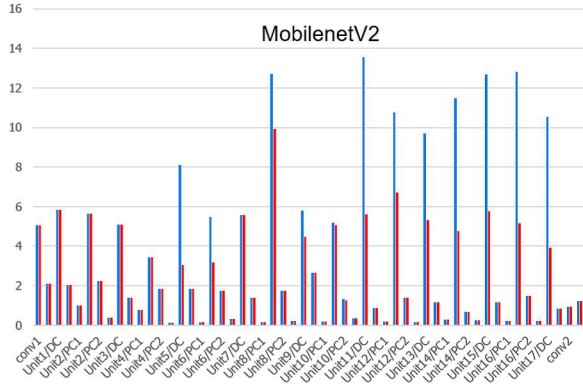

Figure 2: This is a figure comparing the dynamic range of the weights after folding BN with respect to using or not using PfQ. The left figure is about MobileNetV1 and the right one is about MobileNetV2. The blue bin is not using PfQ and the red one is using PfQ. The vertical axis is the dynamic range value (i.e. weight max - weight min), and the horizontal axis is the layer. In each figure of MobileNetV1 and MobileNetV2, the left side on the horizontal axis is the layer on the input side.

From the results in Figure 2, it can be seen that PfQ improves the wide dynamic range.

### 4.2.3 EFFECT OF PROPOSED QUANTIZATION WORKFLOW

In this section, we confirm the effect of Algorithm 1. Table 2 shows the results of comparing the performance between quantization using Algorithm 1 and the fine-tuning of the DNN with activations and weights quantization simultaneously once after performing PfQ. The experiments are the classification task of CIFAR-100 using MobileNetV1 and MobileNetV2. In the each fine-tuning in Algorithm 1, the number of epochs was 100, the optimizer was momentum SGD, the moment was 0.9, the learning rate settings in (19) were $w = 0, \lambda = 100$. Regarding the fine-tuning once with PfQ, the number of epochs is 200 in order to be the same number of the total epochs in Algorithm 1.

Table 2: Effect of proposed quantization workflow

| Model | Bitwidth (Activation/Weight) | Evaluation | fine-tune once (w/ PfQ) | Algorithm 1 |
|---|---|---|---|---|
| MobileNetV1 | A4/W4 | Accuracy | 74.69 | **75.31** |
|  |  | MAC | 367M | 367M |
| MobileNetV1 | A3/W3 | Accuracy | **71.56** | 70.85 |
|  |  | MAC | 367M | 367M |
| MobileNetV2 | A4/W4 | Accuracy | **72.78** | 72.7 |
|  |  | MAC | 93M | **92.4M** |
| MobileNetV2 | A3W3 | Accuracy | 66.29 | **68.96** |
|  |  | MAC | 93M | 93M |

Table 2 shows that Algorithm 1 is better. Although the performance of Algorithm 1 is lower than that of the fine-tuning once with PfQ for A3W3 of MobileNetV1 and A4W4 of MobileNetV2, the difference is small, and Algorithm 1 which can often reduce the MAC is considered to be better.

### 4.3 COMPARISON TO OTHER METHODS

In this section, we apply the proposed quantization workflow using PfQ to the pre-trained model. Here, we compare the performance of the proposed method with that of the other quantization methods. The experimental settings common to each are described below. We used MobileNetV1 and MobileNetV2 as a network architecture. In the each fine-tuning in Algorithm 1, the number of epochs was 100, the optimizer was momentum SGD, the moment was 0.9, the learning rate settings in (19) were $w = 0, \lambda = 100$. In the experiment of DFQ, the number of epochs was 200, the learning rate settings in (19) were $w = 0, \lambda = 200$. The optimizer, the batch size, the initial learning rate $l_{-1}$ and the weight decay settings were the same settings as Table 3 and Table 4.

### 4.3.1 CIFAR-100 CLASSIFICATION EXPERIMENT

In this subsection, we compare the performance of the proposed quantization method with DFQ using fine-tuning in the classification task of CIFAR-100. Next, we explain the experimental settings. We used a single GPU (2080Ti) for the learning in CIFAR-100. The same pre-trained model as subsection 4.2.1 was used. In our method, we used the same settings as subsection 4.2.1 in the first fine-tuning for quantized activations only in Algorithm 1. In the settings of the second fine-tuning for quantized activations and weights in the Algorithm 1, the optimizer was SGD (i.e. the moment was 0), $l_{-1} = 0.0005$, the weight decay was 0.00001. In the settings of DFQ, the optimizer was SGD, $l_{-1} = 0.001$, and the weight decay was 0.0001.
The experimental results are summarized in Table 3, which shows that the performance of the proposed method is better than DFQ with fine-tuning. Also, 2 PfQ operations in the proposed quantization workflow pruned the weights by 36.17% and 34.36% in the A4/W4 and A3/W3 models, respectively. Thus, the number of MACs was also reduced.

### 4.3.2 IMAGENET1000 CLASSIFICATION EXPERIMENT

In this subsection, we compare the performance of the proposed quantization method with the other quantization method in the classification task of imagenet1000. The methods used for comparison are DSQ and PACT which achieve high performance with 4 bits in the activations and weights

Table 3: Top1 Accuracy comparison CIFAR-100

| Model | Bitwidth (Activation/Weight) | Evaluation | DFQ (w/ fine-tune) | Ours |
|---|---|---|---|---|
| MobileNetV1 | A4/W4 | Accuracy | 71.57 | **75.31** |
| | | MAC | 510M | **367M** |
| MobileNetV1 | A3/W3 | Accuracy | 61.58 | **70.85** |
| | | MAC | 510M | **367M** |
| MobileNetV2 | A4/W4 | Accuracy | 72.14 | **72.7** |
| | | MAC | 147.2M | **92.4M** |
| MobileNetV2 | A3/W3 | Accuracy | 68.05 | **68.96** |
| | | MAC | 147.2M | **93.0M** |

for MobileNetV2, and DFQ. Next, the experimental settings are described. The experiment of the imagenet1000 was learned using 4 GPUs (tesla v100 x4), the optimizer was momentum SGD, the moment was 0.9, and the weight decay was 0.00001. The pre-trained model was learned by the following settings. The number of epochs was 120, the batch size was 256, the learning rate settings in (19) were $l_{-1} = 0.256, w = 16, \lambda = 104$. The settings for the fine-tuning of the quantized DNN are described below. In our method, in addition to the above settings, the batch size was 64, and the initial learning rate was $l_{-1} = 0.01$ in the first fine-tuning for quantized activations only in Algorithm 1. In the second fine-tuning for quantized activations and weights in the Algorithm 1, the initial learning rate was $l_{-1} = 0.0001$. In the experimental settings of DFQ, the batch size was 128, the initial learning rate was $l_{-1} = 0.0001$.

Table 4: Top1 Accuracy comparison imagenet1000

| Bitwidth (Activation/Weight) | Evaluation | DFQ (w/ fine-tune) | PACT | DSQ | Ours |
|---|---|---|---|---|---|
| A4/W4 | Accuracy | 57.44 | 61.39 | 64.8 | **67.48** |
| | MAC | 313M | 313M | 313M | **261M** |

The experimental results are summarized in Table 4, which shows that the quantization performance of the proposed method is the best compared to the other quantization methods. In our approach, 2 PfQ operations in the proposed quantization workflow pruned the weights by 10.29%. Thus, the number of MACs was also reduced.

These results show that PfQ and the proposed quantization workflow make it easy to fine-tune the quantized DNN and can make high-performance models even with low bits. On the other hand, in this method, STE is used as the gradient of the quantization function, and the problem of degradation of the gradient due to quantization, which is often raised as a problem in the study of quantization, is not addressed. Therefore, further performance improvement can be expected if a technique to cope with gradient degradation during the fine-tuning of the quantized DNN such as DSQ is used together with the proposal of this paper.

## 5 CONCLUSION

In this paper, we propose PfQ to solve the dynamic range problem in the fine-tuning of the quantized DNN and the problem that the quantization errors tend to accumulate in some bias, and we propose a new quantization workflow including PfQ. Since PfQ is a type of filter pruning, it also has the effect of reducing the number of weights and the computational costs. Moreover, our method is very easy to use because there are no additional hyper-parameters and learning parameters like the conventional pruning and quantization methods.

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

# A APPENDIX

## A.1 EXPERIMENTS USING VALIDATION SET

Algorithm 1 needs the fixed number of epochs in the experiments. So we made the experiments not using the fixed number of epochs and using the value of accuracy drop. The accuracy check during the training used validation set apart from training set and test set. We split a separate validation set out of the original training set by randomly sampling 50 images from the training set for each category. We set the accuracy drop 0.5% in the each fine-tuning in Algorithm 1 and 1.0% in DFQ. Then the end numbers of epochs were 82 and 80 during the each fine-tuning in Algorithm 1 and the end number of epoch of DFQ was 200 that was we set max epochs. The result is Table 5.

Table 5: Using Accuracy Drop

| Model | Bitwidth (Activation/Weight) | Evaluation | DFQ | Ours |
|---|---|---|---|---|
| MobileNetV2 | A4/W4 | Accuracy | 70.00 | **71.58** |
| | | MAC | 147.2M | **92.4M** |

## A.2 VISUALIZE LOW VARIANCE CHANNELS

We claimed in subsection 3.1 that the outputs of some channels whose running variance close to zero are the constant values. In this subsection, we confirmed the fact and the result was Figure 3. Figure 3 shows that the channels whose running variances are close to zero are the constant value. The model we used in Figure 3 was the pre-trained model in subsection 4.2.1. The input images are in the CIFAR-100 training set.

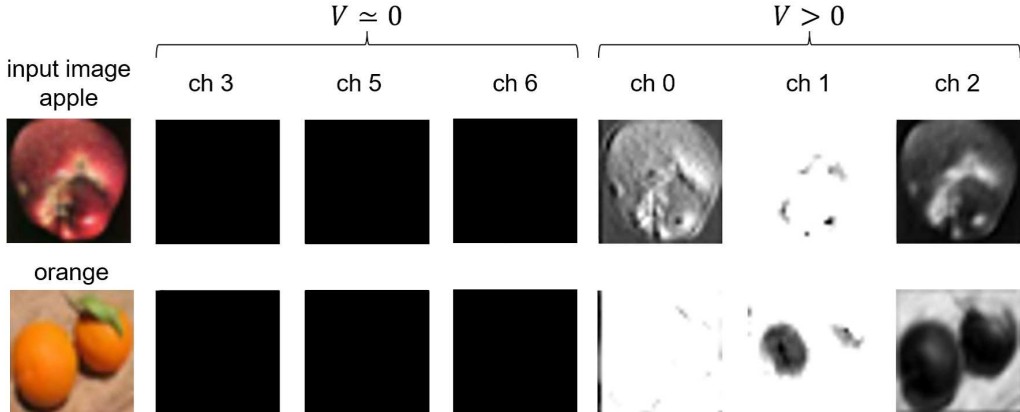

Figure 3: This is the visualized channels of the first BN outputs after the first convolution in MobileNetV2. The input images are apple and orange, respectively. The running variance of the channel 3,5,6 are close to zero and that of the channel 0,1,2 are not close to zero.

