# OpenReview forum: "Filter pre-pruning for improved fine-tuning of quantized deep neural networks"
_ICLR.cc/2021/Conference — Reject_

### Official Review · AnonReviewer4 · 2020-10-25
**Review of Paper 609**

**Rating:** 5
**Confidence:** 3

**Review:**

This paper proposes a pruning for quantization (PfQ) scheme to improve the fine-tuning of the quantized network. PfQ removes the filters that disturb the fine-tuning of the DNN while retaining good performance. Some experiments are conducted to demonstrate the effectiveness of the proposed PfQ.

Pros:
- The paper is well motivated and the method seems valid.

Cons:
- The writing quality is poor. in its current form, a general readership will struggle to understand it. An expert familiar with the details of the field (not just the general area) can probably disentangle the text, but otherwise, it's too convoluted.
- This paper is over-claimed. For example, how BatchNorm disturbs the fine-tuning of the quantized network has been pointed out by [1]. The analysis presented by the authors is too incremental.

Some remarks:
- Please note that BN can not be absorbed into the previous convolution layer in the training stage. The mean and variance in Eq.(2) should be running-moving average statistics, which should be pointed out in the beginning.
- To improve the quality of this paper, more rigorous mathematical notations and some visual experiments should be provided.
- It would be better to show some ablation studies to show how your method works.

[1] Data-free quantization through weight equalization and bias correction. Markus Nagel et al.

**********After rebuttal

The revised version has a better shape. In particular, I like the analytical experiment (Fig.2), which demonstrates that the proposed scheme can improve the wide dynamic range. Overall, this paper observes that BN with small variance influence quantization and proposes a protocol for training a quantized neural network combining filter pruning.

Some issues still prevent it from being accepted. For example, PfQ is proposed to reduce the dynamic range. However, there is even no definition of the dynamic range in the paper which may make readers hard to understand the mechanism of PfQ.  Besides, the author claim that the weight widening the dynamic range in quantization is theoretically analyzed. But the analysis of Eqn. (14-16) is less rigorous. The readers may expect to see how weights in the case of $V_{c}^{L,\tau} \approx 0$ increase the dynamic range according to its definition compared to those weights in the other cases ($V_{c}^{L,\tau} \gg 0$).

The paper proposes an effective approach of quantization, which reduces the model size and improves accuracy. I would like to increase my rating to 5.

---

> ### Author Response · Authors · 2020-11-18
> **Answer to reviewer4**
>
> Thanks for your kind comments. Please find our reply below.
>
> Answer for Cons 1:
>  - We will scrutinize and revice this paper so that readers who are not familiar with this field will understand it.
>    If possible, could you tell me where you felt that?
>
> Answer for Cons 2:
>  - The claim in [1] is about the dynamic range of the weights.
>    In this paper, we claim that BatchNorm has yet another problem described in subsection 3.1 after we point out that the problem of the dynamic range is claimed by [1].
>
> Answer for Some remarks 1:
>  - Thank you for your clear advice.
>    It is described in the sentence under (4) of Subsection 2.2.
>
> Answer for Some remarks 2:
>  - We think this is probably related to Cons 1.
>    We scrutinize and revise this paper.
>    If possible, could you tell me where you felt that?
>
> Answer for Some remarks 3:
>  - As pointed out by other reviewers, we will make the experiments about the ablation studies for workflow.
>    We will also consider if there are any other ablation studies to confirm.
>    What kind of ablation study did you feel you should do?

---

> > ### Author Response · Authors · 2020-11-24
> > **paper updated and we made some experiments**
> >
> > Answer for Some remarks 1:
> >  - We revised some sentence about that to clear below the equation (2) in subsection 2.2.
> >
> > Answer for Some remarks 2:
> >  - We understood as that the one is about the too many indexes in subsection 2.2 and 3.1 and another is that we should visually describe the experiments.
> >    Therefore, we revised some equations in subsection 2.2 and 3.1 to reduce indexes and clear and visually show the results of solving the problem of dynamic range by PfQ in subsection 4.2.2 and some channels output of BN to describe the difference between the running variances in appendix A.2.
> >
> > Answer for Some remarks 3:
> >  - We added the results of ablation study of our quantization workflow in subsection 4.2.3.

---

### Official Review · AnonReviewer2 · 2020-10-26
**Clear idea. More analysis and results are needed to support the idea**

**Rating:** 6
**Confidence:** 3

**Review:**

### Overall
This work present a Pruning mechanism for Quantization scenario. Duo to the low-bits effects, the quantized network is hard to train properly. Therefore, authors provide a new method call Pruning for Quantization (PfQ) and a workflow to solve the model compression problem practically. Comparing to some current quantization methods, PfQ obtains some gain from solution to the performance on benchmark datasets such as ImageNet and CIFAR100.

### Pros
Clear idea about PfQ and the mathematical results provide the reason why the authors want to do so.

### Cons
1) Generally, the joint quantization (Q) and pruning (P) framework is not novel at present. For example, the following papers have tried to solve the model compression problem with P and Q jointly:
Tung, Frederick, and Greg Mori. "Clip-q: Deep network compression learning by in-parallel pruning-quantization." Proceedings of the IEEE Conference on Computer Vision and Pattern Recognition. 2018.
Tung, Frederick, and Greg Mori. "Deep neural network compression by in-parallel pruning-quantization." IEEE transactions on pattern analysis and machine intelligence (2018).
Wang, Ying, Yadong Lu, and Tijmen Blankevoort. "Differentiable Joint Pruning and Quantization for Hardware Efficiency." European Conference on Computer Vision. Springer, Cham, 2020.
And compression with the BN mechanisms are also available:
Gao, Xitong, et al. "Dynamic channel pruning: Feature boosting and suppression." arXiv preprint arXiv:1810.05331 (2018).
Liu, Yuan, et al. "Local Normalization Based BN Layer Pruning." International Conference on Artificial Neural Networks. Springer, Cham, 2019.
Kang, Minsoo, and Bohyung Han. "Operation-Aware Soft Channel Pruning using Differentiable Masks." arXiv preprint arXiv:2007.03938 (2020).
To support the claim that PfQ has the SOTA performance on benchmark datasets, first, PfQ should outperforms other BN based pruning techniques under the quantization settings. Then, PfQ should be better than those Pruning and Quatization Joint optimal methods for model compression. Especially, the readers would like to know why the variance based pruning approach is better than those BN weights based methods. It is not convincible enough for readers to follow the idea without analysis into details.
2) It is lack of explanation why the workflow is needed for the model compression with PfQ. The reason why multiple PfQs are executed is missing. A more clear algorithm would be better for the presentation of the "workflow".
3) Ablation studies: since the author design multiple round of PfQs, the paper is expected to show the studies on what will happen if the workflow just contains PfQ and reasonable finetune. Why the performance could not beat the current "workflow"? How would the variance distribution be after the "workflow"?
4) More proof-reading will make the paper look better. For example,  in page 7, section 4.3.1, "a single GPU (2080ti)" => "a single GPU (2080Ti)". In the same sentence, "for the learning in cifar100" => "for the learning in CIFAR-100".

---

> ### Author Response · Authors · 2020-11-18
> **Essence of this paper is the new problem of quantization (equation (16))**
>
> Thanks for your kind comments. Please find our reply below.
>
> Answer for weakness 1:
>  - we appreciate that you presented several important papers.
>    However, the presented mathods are inappropriate as comparative methods to ours.
>    PfQ is the simplest pre-processing method to solve the problem of quantization that is discussed in this paper (i.e., certain weights disturb the fine-tuning of the quantized DNN).
>    For this reason, quantization methods that do not consider joint quantization and pruning are chosen as our comparative methods.
>    Specifically, we do not discuss the effectiveness of PfQ as pruning.
>    The main contribution of this paper is showing the effectiveness of removing certain weights that disturb the fine-tuning.
>    Therefore, comparing with the methods that consider effective joint quantization and pruning is unfair.
>
>
> Answer for weakness 2:
>  - "It is lack of explanation why the workflow is needed for the model compression with PfQ."
>   PfQ is independent of the workflow.
>     The workflow is proposed to improve the performance after the fine-tuning of the quantized DNN.
>     In general, DNN quantization is performed for the activations and weights simultaneously for float pre-trained model.
>     However, the fine-tuning may not proceed well by this method since the variation from the pre-trained model is too large.
>     To deal with this problem, the proposed workflow is introduced to fine-tune progressively that helps the fine-tuning process.
>     Also, our quantization workflow can get the effect of BN.
>     In general, for DNN quantization, BN layers are folded into the convolution before quantization to reduce the weights that have to be quantized.
>     However, our proposed workflow allow us to use BN during the first fine-tuning since the weights are treated as float values in the first fine-tuning.
>     Therefore, the effect of BN (i.e., the statistical information of the layers can be used) is obtained during the first fine-tuning and improvements in the performance can be expected than the previous methods.
>  - "The reason why multiple PfQs are executed is missing."
>   During the first fine-tuning for quantized activations only in our quantization workflow, the model contains BN layers.
>     This may cause that the filters that disturb the fine-tuning for quantized DNN are newly produced after the first fine-tuning.
>     To solve this problem, the second PfQ is performed to remove these filters.
>  - "A more clear algorithm would be better for the presentation of the "workflow"."
>   We appreciate your precise advice. "Proposal of Quantization Workflow" in Section 3.3 was modified to show a more clear algorithm.
>
> Answer for weakness 3:
>  - "since the author design multiple round of PfQs, the paper is expected to show the studies on what will happen if the workflow just contains PfQ and reasonable finetune. "
>   We appreciate your precise advice.
>     As commented, the ablation study is necessary to show the importance of our proposed workflow.
>     We will make the additional experiments, and write the results in the paper update.
>  - "Why the performance could not beat the current "workflow"?"
>   We will make the experiments as the above ablation study.
>  - "How would the variance distribution be after the "workflow"?"
>   The variance distribution after the "workflow" did not bave any noticeable characteristics as overall.
>     However, there was almost no variance close to zero.
>     For example, the filters close to zero existed only 0.3% in MobileNetV2 after the "workflow" for CIFAR-100.

---

> > ### Author Response · Authors · 2020-11-24
> > **paper updated and we made some experiments**
> >
> > Additional comment for weakness 2:
> >  - "The reason why multiple PfQs are executed is missing."
> >   We revised subsection 3.3 to give more explanation of our quantization workflow.
> >  - "A more clear algorithm would be better for the presentation of the "workflow"."
> >   We revised our paper and the more clear algorithm was described in subsection 3.3.
> >
> > Additional comment for weakness 3:
> >  - We made the ablation study regarding our quantization workflow and the results was described in subsection 4.2.3.

---

### Official Review · AnonReviewer1 · 2020-10-29
**borderline, tend to accept**

**Rating:** 6
**Confidence:** 4

**Review:**

This paper proposed to prune certain channels to improve the accuracy of quantized DNN model. The motivation comes from the observation that the channels which have small variance are actually harmful to the quantization-aware training. The authors show that these channels with small variance can be pruned without significant influence to the neural network, and the accuracy loss can be easily recovered with fine-tuning.

Pros:

This paper makes an important observation that channels with small variance lead to a problem that the filter weights will have a wide range after fusing the BN layer into convolution. Directly performing quantization on the weight tensor will lead to a large quantization scale and large quantization error.

The authors find that the channels with small variance can actually be removed. Based on their following BN operations, these channels can be seen as constant channels approximately, and these constants can be folded into the bias of the next layer. By this way, the neural network is not only more suitable to quantize, but also has less model size and computation.

The paper gives a study showing that removing the channels with small variance improves the accuracy of quantized model. The proposed method is also compared with several other quantization methods.

Cons:

On ImageNet, the experiments are only did on MobileNetV2, and only a few methods are compared. Although the current experiments show the benefits of PfQ, it is better to have more experiments to make the validation part more solid.

Table 1 is to show the benefits of removing disturbing weights. Why use the activation quantization instead of weights quantization? It's better to analyze weight quantization here too.

There are some questions which are not clear in the paper:

1. Will per-channel quantization have the problem discussed in this paper? What about the results of using / not using PfQ for per-channel quantization? The min-max quantization scale (eq 20) is used in this paper, what if we use learnable quantization (e.g., LSQ [1])? Is it still suffer from the channels having small variance?

2. DFQ is used as the quantization method. Is this a quantization method cited in this paper? I didn't find the reference and the full name for DFQ.

3. Is the proposed method only applied to DNNs which have BN layers? Is it possible to apply the proposed method to models without BN layers?

Other comments:
It's easier for people to understand the key idea of this paper if moving Figure 1 to the first few pages.

In general I think this paper is a good work on quantization and model compression. It makes good observation of factors that influence quantization, and shows that pruning can actually help quantization in some cases. There are some issues on the experiment (See cons), it's better to validate the proposed method on more DNN architectures and compare with more quantization methods.

References

[1] Esser, S.K., McKinstry, J.L., Bablani, D., Appuswamy, R. and Modha, D.S., 2019. Learned step size quantization. arXiv preprint arXiv:1902.08153.

---

> ### Author Response · Authors · 2020-11-18
> **Answer to reviewer 1**
>
> Thanks for your kind comments. Please find our reply below.
>
> Answer for your comment 'Table 1 is to show the benefits of removing disturbing weights. Why use the activation quantization instead of weights quantization? It's better to analyze weight quantization here too.':
> There is a reason why we did not experiment with weight quantization.
> A filter with a small variance of BN causes two problems:
> problem (1) that increases the dynamic range of the weights and problem (2) that disturbs the fine-tuning of the quantized DNN.
> Problem (1) is claimed in [1] below, and problem (2) is our claim.
> When we fine-tune the DNN with quantized weights, problem (1) must be solved by some way. (Otherwise, the fine-tuning may not work.)
> Therefore, in the experiment in Table 1, we made the experiment with only activation quantization in order to focus only on problem (2).
> [1]: A Quantization-Friendly Separable Convolution for MobileNets (https://arxiv.org/abs/1803.08607)
>
> Answer for question 1:
> - "Will per-channel quantization have the problem discussed in this paper?"
>   It is theoretically possible.
>   The above problem (2) is caused by the quantization error and gradient approximation error of the second and fourth terms of equation (16).
>   By per-channel quantization, if the errors are very small, the problem may not occur, but if the errors become large due to low-bit quantization, etc., the problem may occur.
> - "What about the results of using / not using PfQ for per-channel quantization?"
>   It works, but the benefits are minor compared to per-layers.
> - "The min-max quantization scale (eq 20) is used in this paper, what if we use learnable quantization (e.g., LSQ [1])? Is it still suffer from the channels having small variance?"
>   The performance may be improved.
>   However, it is likely that the performance will be lower than the case using PfQ, because the training will also have to work to the problem of small channel variance.
>
> Answer for question 2:
>   The following [2] is cited. We forgot to add a link. I'm so sorry.
>   [2]: Data-Free Quantization Through Weight Equalization and Bias Correction (https://arxiv.org/abs/1906.04721)
>
> Answer for question 3:
>   Our method currently applies only to DNNs with BN layer. How to apply it to a model without BN layer has not yet been considered.
>   The essence of the problem claimed in this paper is that some channels whose values do not change for any input data in the model disturb the quantization fine-tuning.
>   Therefore, we consider that the essential claim of this paper can be applied if such channels can be detected without BN.
>
> Answer for other comments:
>   Thank you your kind comments. We will move the Figure 1 just before Section 2.

---

> > ### Author Response · Authors · 2020-11-24
> > **paper updated**
> >
> > We update our paper.
> > The changes are commented above.
> > Could you check our paper.

---

### Official Review · AnonReviewer3 · 2020-10-30
**Effective idea with limited novelty**

**Rating:** 5
**Confidence:** 3

**Review:**

This paper studies the effect of quantization during training together with batch normalization in quantized deep neural networks. The compound effect of convolution and batch normalization on the dynamic range of activations has implications on the progress of training. The authors propose a protocol for training a quantized neural network combining filter pruning, fine tuning and bias correction. The experiments show that the model size is reduced significantly, while keeping or improving the accuracy.

Strengths
- The paper is clearly presented, and the effect of batch normalization in the dynamic range of a quantize layer is interesting.
- The method, in the paper settings, is effective in reducing the model size and sometimes improving accuracy.

Weaknesses
- There is not qualitative analysis of the problem of the dynamic range and how the proposed method alleviates it.
- The novelty is limited in my opinion. The paper combines typical practices used in other works. For example, pruning filters with small magnitude is a common practice. it is also common to fine tune a network after pruning or compression, which not only reduces model size but often also improves the accuracy due to lower overfitting. And bias correction to account for the reconstruction error is also commonly used in similar cases (e.g. DFQ, Finkelstein2019, Masana2017).
- The number of iterations in the experiments is a fixed number. It would be more convincing using a validation set with early stopping.
- The pruning+fine tuning seems to be done once (twice in the proposed workflow). In that case the comparison may not be fair without several iterations of pruning+fine tuning, to compare models when they cannot be further pruned.

Overall, I think the novelty of the paper is limited, with concerns about the experiments.

Questions
Please address weaknesses

Finkelstein et al., Fighting Quantization Bias With Bias, ECV@CVPR 019
Masana et al., Domain-adaptive deep network compression, ICCV2017

-- Post rebuttal

I appreciate the response by the authors and the new experiments. I also read the other reviews and responses. I think the paper has improved in the revised version. However, I'm still concerned about the novelty, which still remains relatively incremental, as also pointed by other reviewers. I update my rating to 5.

---

> ### Author Response · Authors · 2020-11-18
> **Our main novelity is to find and solve the new problem of quantization (equation (16))**
>
> Thanks for your kind comments. Please find our reply below.
>
> Answer for weakness 1:
> We completed some experiments which show that PfQ suppresses the dynamic range of the weights.
> We will add these results in the update of our paper.
>
> Answer for weakness 2:
> This paper claims a new problem in quantization (see Equation (16) and its explanation in section 3.1) and the simplest solution to solve it and improve the performance.
> So it's not just a combination of the traditional methods.
>
> Answer for weakness 3:
> Thank you for your helpful suggestions. We will do some experiments to make our paper more convincing. We will include these additional results in the update of our paper.
>
> Answer for weakness 4:
> The fine-tuning in our paper is done for quantized activations and weights, not for pruning. Therefore, the number of pruning does not lose fairness.

---

> > ### Author Response · Authors · 2020-11-24
> > **paper updated**
> >
> > We made the new experiments and the results were described in subsection 4.2.2 for weakness 1 and appendix A.1 for weakness 3.
> > Could you confirm the results.

---

### Author Response · Authors · 2020-11-24
**Paper updated**


Thank you for your comments and reading our paper.
We update the paper.
This update contains the following changes.
 - Revising the structure of subsection 4.2.
   The title of 4.2 is changed to "Ablation Study" and subsection 4.2.1 is "Effect of Disturbing Weights" (i.e. old 4.2), 4.2.2 is a new subsection "Solving Dynamic Range Problem" and 4.2.3 is a new subsection "Effect of Proposed Quantization Workflow".
 - Addition of Appendix about the two additional experiments.
   The first result is regarding early stopping using validation set.
   The second result is visualizing the difference between the channels whose running variance is close to zero and the channels whose running variance is not close to zero.
 - Revising the description of our quantization workflow in subsection 3.3.
 - Simplifying some mathematical notations in subsection 2.2, 3.1, 3.2.
 - Addition of the additional results in Table 3 (old Table 2) about MobileNetV1.
 - Moving figure 1 above section 2.
 - Addition of hyperlinks in each references.

---

### Decision · Program_Chairs · 2021-01-07
**Final Decision**

**Decision:**

Reject

**Comment:**

Four reviewers rate this article borderline. R3 finds the paper clearly presented and the method effective, but misses quantitative analysis of the dynamic range problem as well as novelty. Following the discussion and revision, she/he considers the paper improved and updated the score to 5, still being concerned about the novelty. R1 considers the paper makes an important observation but has concerns about experiments, rating it 6. R2 considers that the paper contributes a clear idea, but indicates that more analysis and supporting results are needed. She/he indicated a number of shortcomings in the initial review, and found the update good, hence tending to rate the paper higher after the responses, 6. R4 considers the paper well motivated and the method valid. However, he/she found the writing poor and over-claiming results, and that more rigorous mathematical notation would help. After the discussion and revision, he/she found the paper better and increased the score to 5, but still found issues preventing the paper from being accepted. In summary, the reviewers agree that the paper contains an interesting and well motivated method, but they also point at a number of shortcomings. The revision improved several of them but others persisted. Although the ratings improved after the discussion, the overall rating is borderline. This is a very competitive call, and hence I have to recommend reject at this time.